# Oil Media on Paper: Investigating the Effect of Linseed Oils on Lignocellulosic Paper Supports

Penelope Banou [1,*], Stamatis Boyatzis [1], Konstantinos Choulis [1], Thanasis Karabotsos [1], Dimitris Tsimogiannis [2], Lamprini-Areti Tsakanika [2], Constantina Tzia [2] and Athena Alexopoulou [1]

[1] Department of Antiquities and Works of Art, University of West Attica, Ag. Spyridonos Str., 12243 Egaleo, Greece; sboyatzis@uniwa.gr (S.B.); kchoulis@uniwa.gr (K.C.); akarab@uniwa.gr (T.K.); athfrt@uniwa.gr (A.A.)

[2] School of Chemical Engineering, National Technical University of Athens, Zografou Campus, 9 Iroon Polytechniou Str., 15780 Zografou, Greece; ditsimog@chemeng.ntua.gr (D.T.); btsakanika@gmail.com (L.-A.T.); tzia@chemeng.ntua.gr (C.T.)

* Correspondence: pbanou@uniwa.gr; Tel.: +30-6977218522

**Abstract:** Condition assessment of works of art created with oil media on paper could be a complex matter when presenting problems of damage due to the absorption of oil binders by the paper support, since they depend on several factors and occur in variable conditions. The present work refers to the results of an investigation on the effect of linseed oils on the color, opacity, morphology, tensile strength, and chemical properties of lignocellulosic papers, in comparison to that of pure cellulosic papers. Lignocellulosic papers are involved in research on new, yet significant, parameters that might influence the behavior of the oil-impregnated areas of the supports upon aging. The research was applied to mock-ups, made of two types of lignocellulosic paper impregnated with three types of linseed oil and subjected to accelaratated ageing in specific conditions of relative humidity and temperature in closed environment. The research involved colorimetry, opacity, tensile strength, pH measurements, SEM, FTIR, and VOC analysis with GC-MS. The results indicated that thermal-humid ageing caused the gradual darkening of the oil-impregnated mock-ups, alterations in opacity, and decrease of pH values, depending mainly on the formulation of linseed oil, as well as a reduction in tensile strength. FTIR analysis results indicated that the chemical changes that occur upon ageing supported the recorded optical and mechanical alterations, while VOC emissions are both associated with the paper type and the kinetics of degradation of the different types of linseed oil.

**Keywords:** linseed oil; pure cellulosic paper; FTIR; VOC; GC-MS; colorimetry; opacity; tensile strength; SEM; pH

## 1. Introduction

Oil media on paper, such as oil paintings and oil sketches on paper and prints, but also archival material and books, present alterations on their supports, considered as damage by conservators, which are associated with the effect of oil binders contained in oil colors and traditional oil-based printing inks when absorbed by the paper support [1–3]. Alterations include discoloration, a decrease in pH value, loss of mechanical strength and embrittlement, and cracks and losses in the oiled areas of the paper support [1–6]. The intensity and extent of this alteration/damage determine the condition of a work and conservation decision-making. However, it occurs irregularly, varying from limited and local to overall, as it depends on several parameters, such as the materials (type of paper, oil medium, and pigments) and techniques (e.g., impasto or diluted with solvents or oil) used in the creation of the works [6]. This fact often makes the evaluation of the condition of a collection a composite matter. It should be mentioned that damage to works without priming or a preparation layer on the paper support, or coated papers, has been recorded.

A small number of research works on this matter have published in the past, though they do not provide adequate results to compensate to the interpretation of alterations and problems recorded in original works [7–10]. Recently, the authors of the present work reported the initial results of an investigation into the effect of linseed oils on pure cellulosic paper [11]. Specifically, we have researched the changes in the optical, morphological, mechanical, and chemical properties of pure cellulosic paper mock-ups impregnated with three types of linseed oil, subjected to thermal-humid ageing in air-tight vessels. The study introduced a holistic methodology to study the effect of linseed oil on pure cotton paper supports for the first time, with interesting results for paper conservators to support condition assessment. Three formulations of linseed oil were selected for the preparation of mock-ups, as linseed oil was the most representative oil medium used in oil painting, printing, and typography for over five centuries (from the 15th century AD up to the 20th century) [12–15].

The current work will present the results of the application of the same methodology for the investigation of linseed oils in lignocellulosic paper supports, a significant category of paper. In the 19th century, wood pulp was introduced to industrial papermaking, which became widely used after 1850 to produce several types of commercially available lignocellulosic paper supports for writing, drawing, painting, printing, and printmaking, still in use today [16]. The method of wood-pulp processing and the fibre and pulp content provide additional factors that might influence the changes of the paper–oil system upon ageing.

A comparative study of the results could provide scientific evidence for the effect of linseed oils both on pure cellulosic and lignocellulosic papers upon ageing, of substantial value for the condition assessment of works. The research aims to study the alteration/damagesthat occurs on paper supports due to the absorption of oil binders upon ageing and determine the possible stages of deterioration.

## 2. Materials and Methods

### 2.1. Materials

Three formulations of linseed oil were selected for the preparation of mock-ups: Cold-pressed linseed oil, alkaline refined linseed oil, and stand-oil (Windsor and Newton), which are the same ones used for pure cellulosic papers. The difference in the methodology of oil manufacture provides these formulations with different physicochemical properties, such as wetting power, drying rate, yellowing or darkening, viscosity, rheology, acid value, and degree of polymerization [13,14,17,18].

Mock-ups were made of two types of paper with different lignin content. A typical watercolor paper, Canson® Montval®, white color, cold-pressed, acid-free, made of wood pulp (soft and hardwood fibres) and limited lignin content, without optical brightness additives, 185 gsm (complying with ISO Standard 9706 requirements for permanence (Montval® | Canson® https://en.canson.com/watercolour/canson-montval, accessed on 25 May 2022) (Art & Hobby, Athens, Greece) and a wrapping paper, Kraft paper (Dionisopoulos, paper distributor, Greece), brown color, 100 gsm, buffered, made of soft and hardwood fibres, containing lignin, fillers, additives, and metallic contamination. Application of optical microscopy, EDX analysis, and FTIR analysis provided data on the fiber and pulp content.

### 2.2. Preparation of Mock-Ups and Artificial Ageing

Oil-impregnated mock-ups were air-dried at room conditions of 22 °C and 52% RH for 40 days, hanged in racks using pure cotton threads. Mock-ups were prepared and subjected to accelarated ageing in air-tight vessels in conditions of 77% relative humidity and 80° temperature, the same used for pure cellulosic papers [11].

*2.3. Methods*

The research followed the exact methodology applied to the investigation on pure cellulosic paper supports reported in previous work carried out by the authors of this work [11]. Experimental work included colorimetry, opacity, tensile strength, pH measurements, SEM, FTIR, and VOC analysis with GC-MS.

## 3. Results and Discussion

*3.1. Color Changes*

After 40 days of drying, oil-impregnated white Montval mock-ups take on a yellowish hue, while light brown Kraft mock-ups appear darker. For oil-impregnated Montval mock-ups, it is evident that the intensity of the yellow hue varies among the three sets (M + CP, M + RFm and M + Stl) (Figure 1). This fact has also confirmed that oil processing during manufacture influence the hue of the formulations of linseed oil.

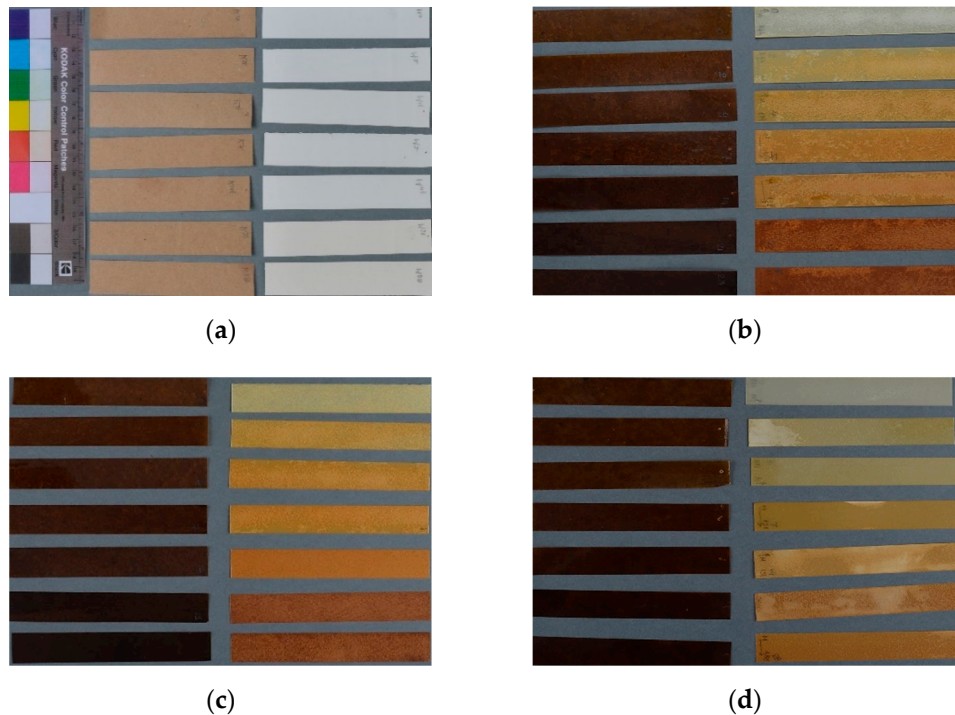

**Figure 1.** Visible light photography: (**a**) sets of plain mock-ups, Kraft (**left**) and Montval (**right**); (**b**) sets impregnated with cold-pressed linseed, K + CP (**left**), and M + CP (**right**); (**c**) sets impregnated with refined linseed oil, K + RF (**left**) and M + RF (**right**); (**d**) sets impregnated with stand-oil, K + StL (**left**), and M + Stl (**right**).

Accelarated ageing resulted in limited changes to plain Montval mock-ups, mainly discernable in those subjected to 21 and 28 days of ageing, while plain Kraft mock-ups presented notable changes after 14 days of ageing (Figure 1). The visible observations were confirmed by ΔE* difference measurements, as the values of plain Montval mock-ups were lower than 2 for those subjected up to 14 days of ageing (the minimal detectable difference is between 1 and 2 ΔE [19,20]), and higher than 5 for those subjected to 21 and 28 days. The values of plain Kraft became higher than 2 for those subjected to artificial ageing for 14 and 21 days, through the value of 3 for those of 28 days. Color changes upon ageing could be attributed to the oxidation of paper in the conditions of accelarated ageing, especially in the final stages (21st and 28th day).

For oil-impregnated mock-ups, thermal-humid conditions of ageing resulted in a gradual darkening, both for Montval and Kraft, with the first to present more distinct variations upon ageing (Figure 1). Oil-impregnated sets of Kraft mock-ups appeared to be darker after 14 days of ageing (Figure 1). However, Montval mock-ups impregnated with

stand-oil (M + Stl) presented color irregularities, possibly due to the uneven oil distribution over the paper surface because of stand-oil's high density and viscosity, as well as limited changes between the ageing stages, a fact that could be attributed to oil processing during manufacture [13], p. 41. In addition, they appeared to be lighter than the other two sets at all stages of ageing, while the mock-ups impregnated with refined linseed oil (M + RF) were darker than those impregnated with cold-pressed linseed oil (M + CP). This observation also applied to the white cotton mock-ups [11]. Oil treatment during manufacture could cause differences in yellowing and darkening upon ageing [17].

Visible observations on plain and oil-impregnated Montval and Kraft mock-ups were supported by the reflectance spectra (Figures 2 and 3) and color difference ΔE* values upon ageing (Figures 4 and 5). The trend of changes of all sets of oil-impregnated Montval mock-ups is quite similar with few variations.

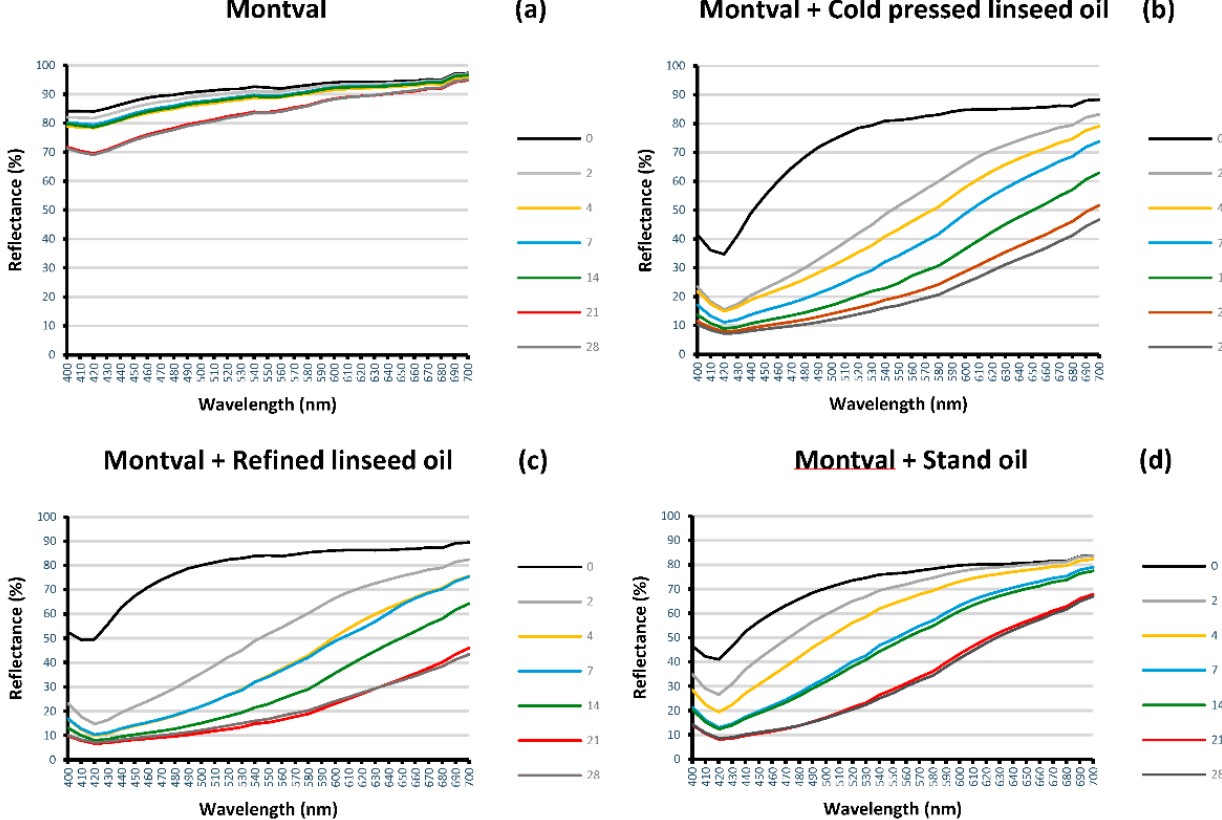

**Figure 2.** Reflectance spectra of all sets of Montval mock-ups: (**a**) plain ones; (**b**) impregnated with cold-pressed oil, M + CP; (**c**) impregnated with refined linseed oil, M + RF; (**d**) impregnated with stand-oil, M + Stl. Measurements of mock-ups subjected to ageing for 0, 2, 4, 7, 14, 21, and 28 days.

The reflectance spectra of the three sets, which were derived from the average values, indicated differences in color changes among the ageing stages (Figure 2). For Montval mock-ups impregnated with cold-pressed linseed oil (M + CP), reflectance spectra indicated gradual darkening with distinct stages, while for those impregnated with refined linseed oil (M + RF) and stand-oil (M + Stl), darkening appeared to be irregular. Color changes could be mainly attributed to oil changes upon ageing. Treatment of oil during manufacture influences the drying rate and consequently the chemical changes associated with color changes [17]. Differences in color changes between the three sets at the various ageing stages could be attributed to the slower rate of drying of refined linseed oil and stand-oil than that of cold-pressed oil [13,17]. Limited changes were recorded for the mock-ups impregnated with refined linseed oil (M + RF), presenting limited changes between the 4th and the 7th day as well as between the 21st and the 28th day, while there was a more intense change between the 2nd and the 4th day than those recorded on the other two

sets. In comparison, those impregnated with stand-oil (M + Stl) presented smaller changes between the 7th and the 14th day, as well as between the 21st and the 28th day, while a more intense change was recorded between the 4th and 7th day. Graphic representation of ΔE* values upon ageing showed a common trend between the three sets, indicating the differences in color changes (Figure 4).

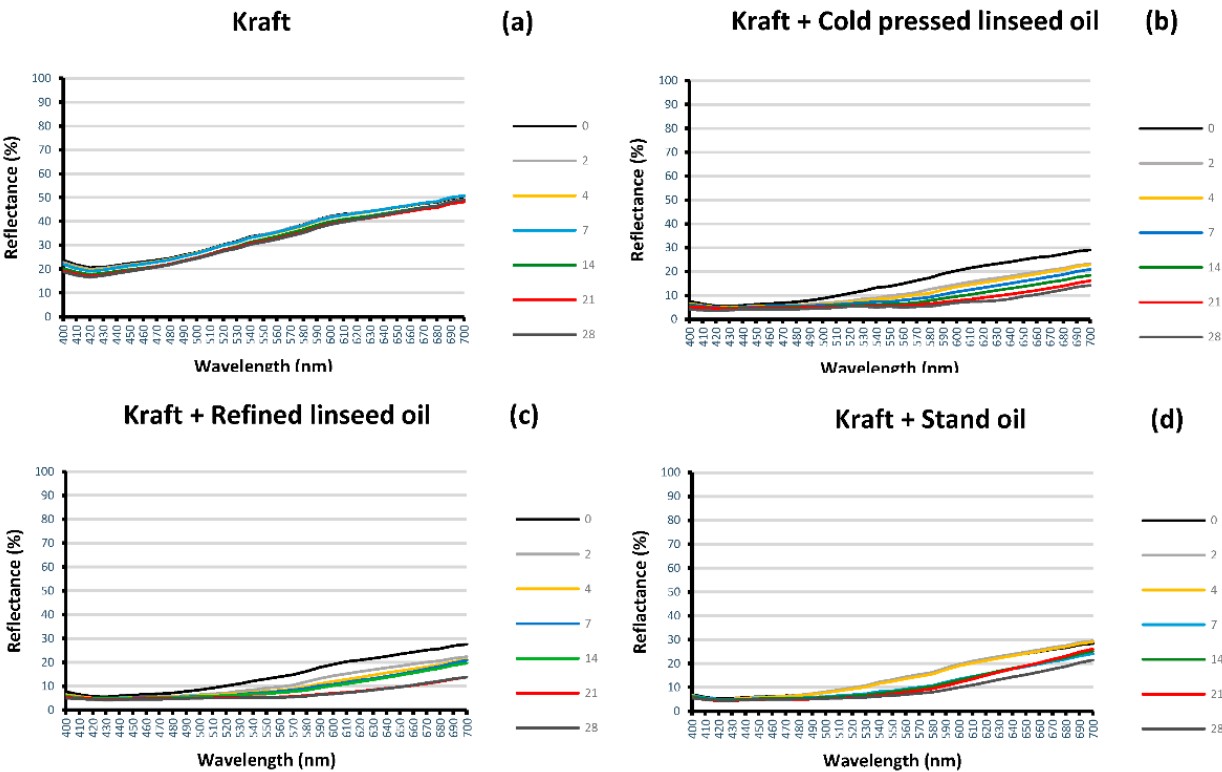

**Figure 3.** Reflectance spectra of all sets of Kraft mock-ups: (**a**) plain ones; (**b**) impregnated with cold-pressed oil, K + CP; (**c**) impregnated with refined linseed oil, K + RF; (**d**) impregnated with stand-oil, K + StL. Measurements of mock-ups subjected to ageing for 0, 2, 4, 7, 14, 21, and 28 days.

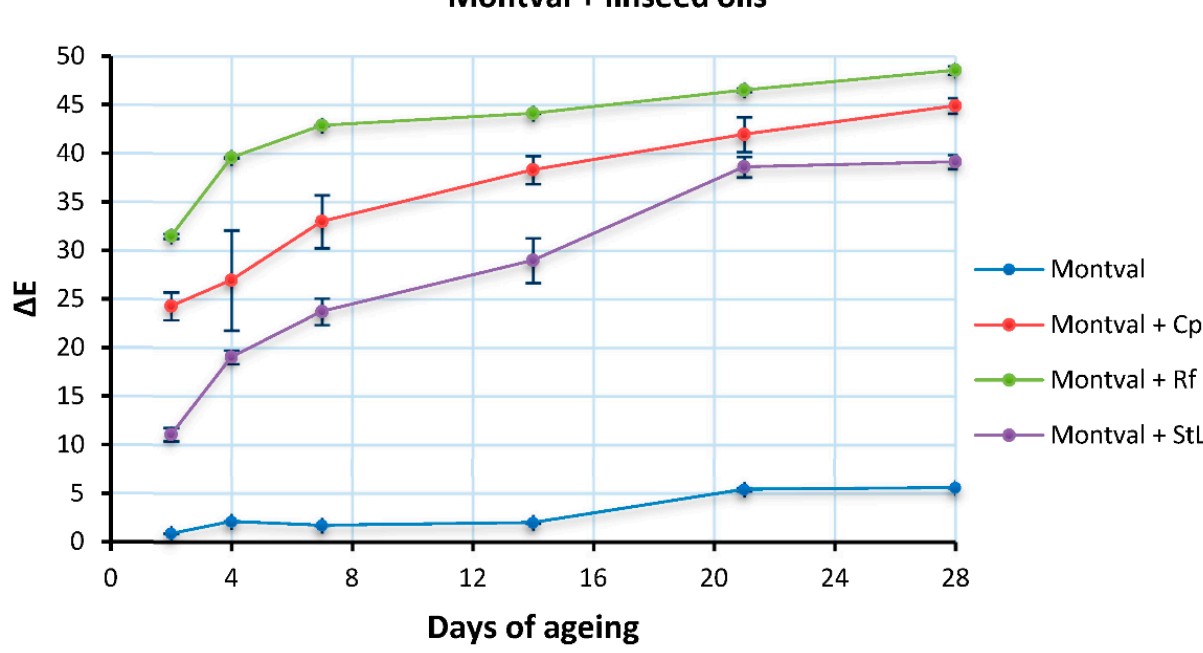

**Figure 4.** ΔE changes of all sets of Montval mock-ups upon ageing.

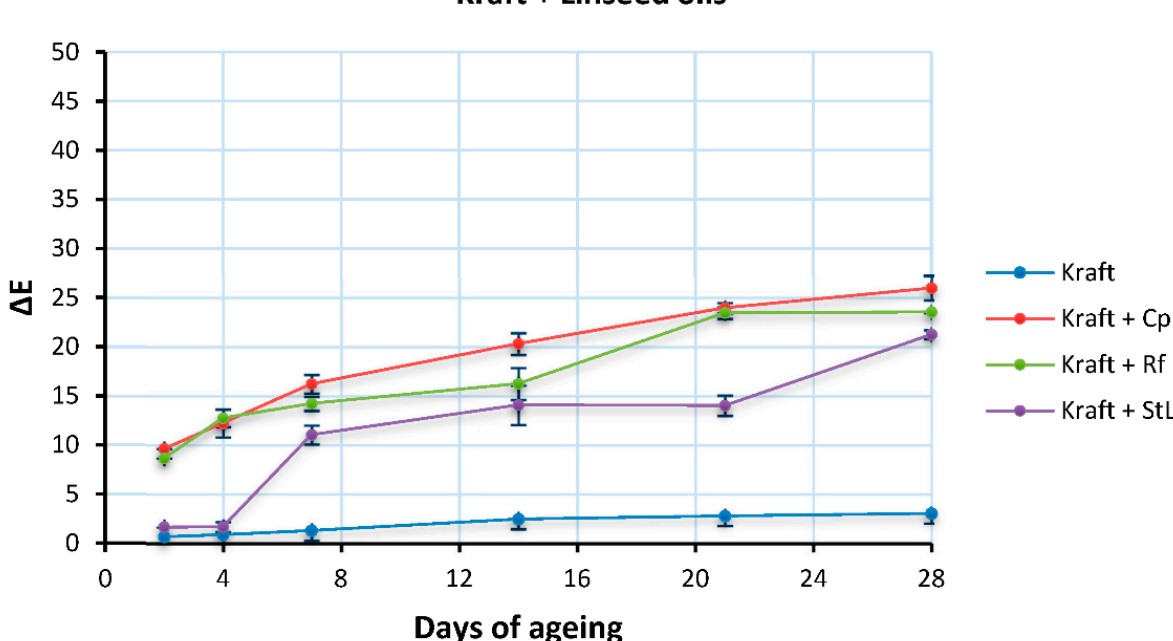

**Figure 5.** ΔE changes of all sets of Kraft mock-ups upon ageing.

Likewise, the reflectance spectra of the oil-impregnated Kraft sets presented differences in color changes among the ageing stages (Figure 3). For Kraft mock-ups impregnated with cold-pressed linseed oil (K + CP), a gradual darkening with distinct stages was indicated, while those impregnated with refined linseed oil (K + RF) presented limited changes between the 4th and 14th day, as well as between the 21st and the 28th day. On the other hand, those impregnated with stand-oil presented insignificant changes up to the 4th day, an intense change until the 7th day, limited changes between the 7th, 14th, and 21st day, and a distinct change until the 28th day. Again, the differences could be attributed to the difference in properties of the three formulations, and in particular, in the drying rate and yellowing/darkening.

Consequently, differences could be due to the rate of oxidation and degradation of each formulation of linseed oil. The standard deviation of ΔE* values confirmed the abovementioned and indicated a common trend for Kraft mock-ups impregnated with cold-pressed linseed oil and refined linseed oil, while for those impregnated with stand-oil, a smaller number of changing stages were indicated (Figure 5). However, the changes of the oil-impregnated Kraft mock-ups upon ageing and between ageing stages were comparatively limited in comparison to those of the oil-impregnated cotton and Montval mock-ups, indicating that Kraft mock-ups were more opaque.

The trend of color changes of oil-impregnated Kraft and Montval papers were similar to those of pure cotton paper in the means of darkening upon ageing, indicating that oil is the principal factor in color changes, while variations depend mainly on oil treatment of formulations. However, the differences in color changes upon ageing between oil-impregnated cotton, Montval, and Kraft mock-ups could be attributed to the color of the plain papers, pulp content and oil penetration, as well as to paper weight (ratio of paper and oil volume).

*3.2. Opacity Changes*

Both plain Montval and Kraft mock-ups presented insignificant changes in opacity upon ageing (increase of opacity of up to 2%). These results indicated that artificial ageing has a limited effect on the opacity of the plain papers. On the contrary, application of the three formulations of linseed oil to Montval and Kraft mock-ups, after 40 days of drying, resulted in a notable reduction of the opacity of the oil-impregnated mock-ups, but to a

different extent (Figures 6 and 7). In particular, the opacity of M + CP mock-ups reduced by 26%, M + RF mock-ups by 35%, and M + Stl by 48%, while K + CP mock-ups reduced by 39%, K + RF mock-ups by 38%, and K + StL mock-ups by 43%.

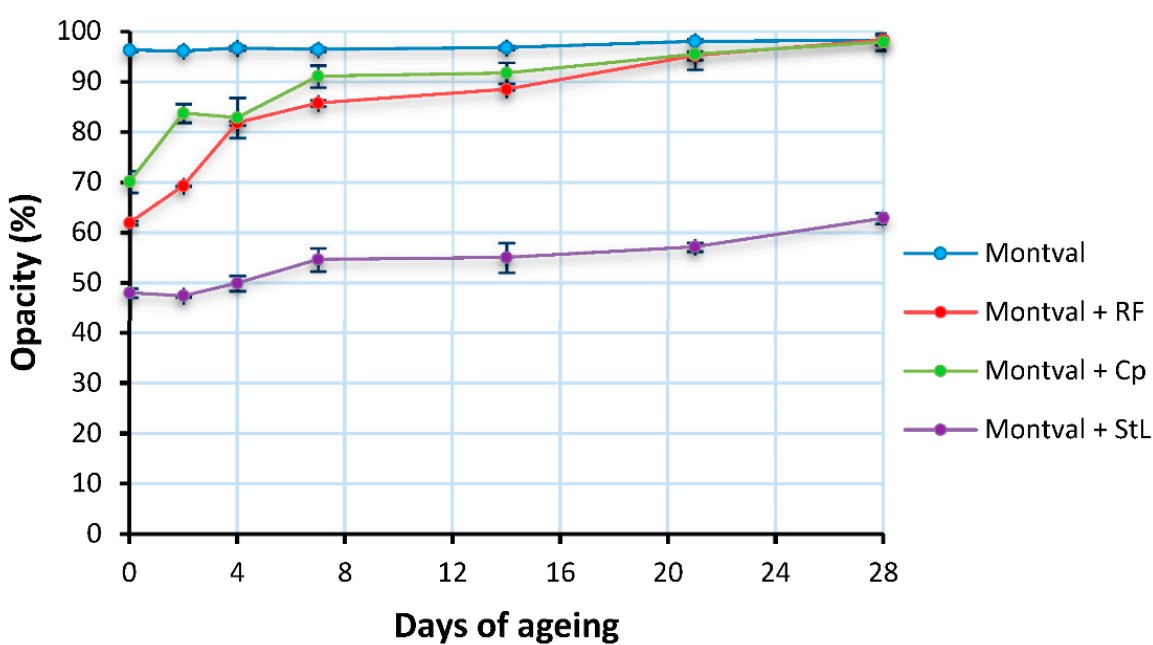

**Figure 6.** Opacity changes of all sets of Montval mock-ups upon ageing.

**Figure 7.** Opacity changes of all sets of Kraft mock-ups upon ageing.

However, application of linseed oils caused an increase in the opacity of the mock-ups upon ageing, reaching up to 48% at the final stages of ageing, with variations following the formulations of linseed oil. Those impregnated with cold-pressed linseed oil and refined linseed oil showed similar trends of change in opacity upon ageing. The opacity of Montval

with cold-pressed linseed oil and refined linseed oil (M + CP and M + RF) increased quite rapidly up to 7 days of ageing, and then to a milder pace up to 28 days of ageing. The trend was quite similar for Kraft mock-ups (K + CP and K + RF), but the changes were more gradual.

In comparison, the range of changes in the opacity of mock-ups impregnated with stand-oil (M + Stl and K + StL) were smaller, as the opacity of the M + Stl mock-ups increased up to 14% and that of K + StL mock-ups up to 25% by the final stages of ageing. Specifically, the opacity of the mock-ups remained almost stable for up to the 4 days of ageing, increased on the 7th day and remained almost stable up to the 21st day, and then presented a notable increase until the 28th day of ageing. Processing of stand-oil during manufacture results in pre-polymerization and high density, which are possibly responsible for the differences in the opacity upon ageing.

It could be suggested that changes in the opacity of the mock-ups can be mainly attributed to linseed oil formulations and their different properties. Taking into consideration the changes in opacity on the oil-impregnated mock-ups of cotton [11], which has similar weight to Kraft, results show that oil-impregnated Kraft mock-ups are less opaque, indicating differences in linseed oil's penetration into the paper pulp. This also confirms the observations in color changes (Section 3.1). On the other hand, the weight difference of Montval paper, and differences in the ratio of paper and oil volume respectively, had an effect on the trend of changes in opacity in the various ageing stages.

*3.3. Morphological Changes*

SEM combined with EDX mapping indicated the presence of alkaline buffer for plain Montval and Kraft papers, but also the presence of fillers and impurities for Kraft paper. The particles of $CaCO_3$ were spread throughout the fiber net of Montval paper without filling out the space/voids between the fibers (Figure 8). On the other hand, the fiber net of Kraft paper was loaded with fillers, additives, and impurities, creating a dense mass (Figure 9).

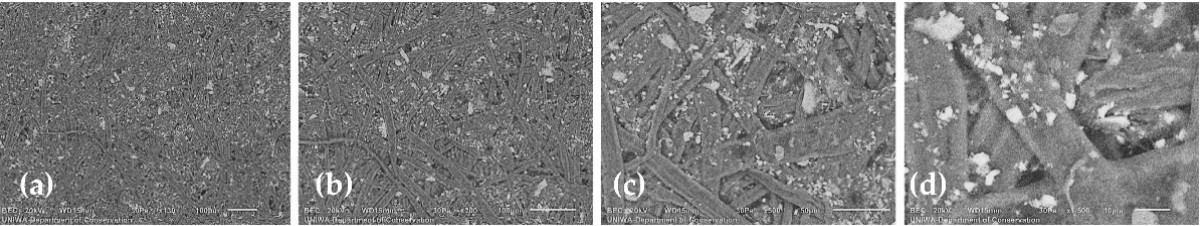

**Figure 8.** SEM images of plain Montval mock-ups at 0 days of ageing: (**a**) 130× magnification; (**b**) 200× magnification; (**c**) 500× magnification; and (**d**) 1000× magnification.

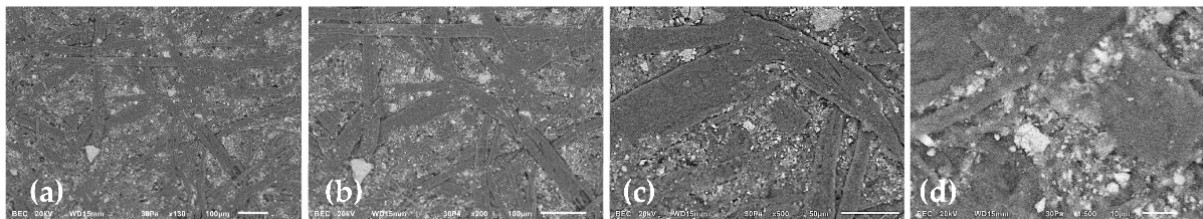

**Figure 9.** SEM images of plain Kraft mock-ups at 0 days of ageing: (**a**) 100× magnification; (**b**) 200× magnification; (**c**) 500× magnification; and (**d**) 1000× magnification.

After 40 days of air-drying of oil-impregnated mock-ups, SEM images showed that a gelatinous film covered the surface of Montval and Kraft mock-ups following the relief of the pulp/fiber net (Figures 10–13). When the liquid linseed oil turns to a semi-elastic gel upon drying, it appears to fill the fiber net. Upon ageing, the oil film recesses gradually while holes open locally, and, by the final stages, the fibers became exposed and particles

of fillers and additives evident. Oil impregnated mock-ups with cold-pressed and refined linseed oil presented similar changes for both papers, providing similar images. In comparison, images indicated that recess of the highly viscous stand-oil was comparatively limited upon ageing, without exposing the fiber net at the final stages Figures 11 and 13).

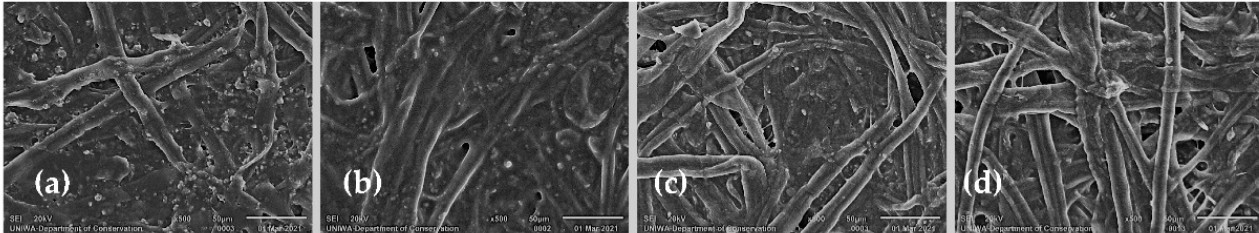

**Figure 10.** SEM images of cold-pressed linseed-oil-impregnated Montval mock-ups upon ageing, at 500× magnification: (**a**) after air-drying for 40 days; (**b**) subjected to 7 days of ageing; (**c**) subjected to 14 days of ageing; (**d**) subjected for 28 days of ageing.

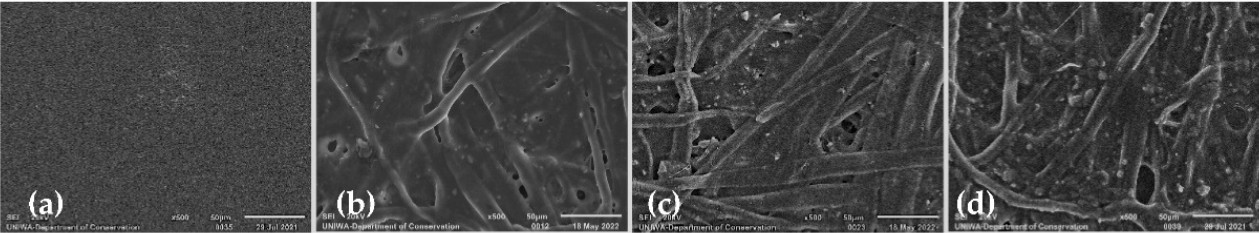

**Figure 11.** SEM images of stand-oil-impregnated Montval mock-ups upon ageing, at 500× magnification: (**a**) after air-drying for 40 days; (**b**) subjected to 7 days of ageing; (**c**) subjected to 14 days of ageing; (**d**) subjected for 28 days of ageing.

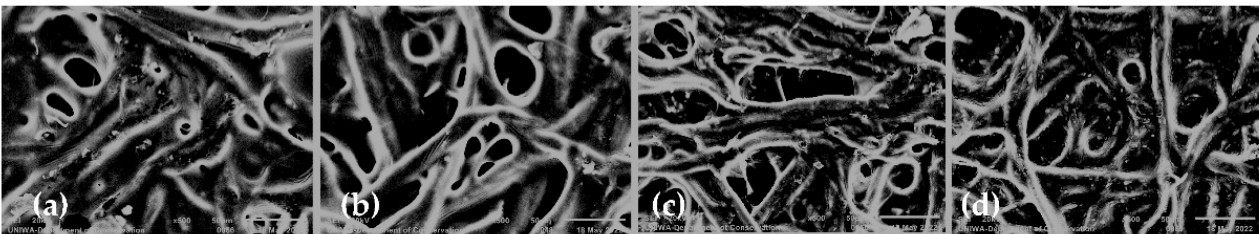

**Figure 12.** SEM images of cold-pressed linseed-oil-impregnated Kraft mock-ups upon ageing, at 500× magnification: (**a**) after air-drying for 40 days; (**b**) subjected to 7 days of ageing; (**c**) subjected to 14 days of ageing; (**d**) subjected for 28 days of ageing.

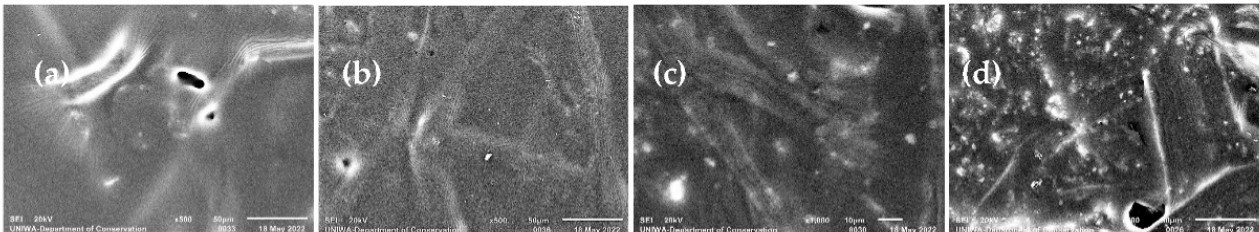

**Figure 13.** SEM images of stand-oil-impregnated Kraft mock-ups upon ageing at 500× magnification: (**a**) after air-drying for 40 days; (**b**) subjected to 7 days of ageing; (**c**) subjected to 14 days of ageing; (**d**) subjected for 28 days of ageing.

This confirms the observations in the opacity changes of oil-impregnated mock-ups with stand-oil (Section 3.2). However, on the Kraft mock-ups, the oil film recess is even less than on the Montval ones (Figures 11 and 13), as well as the cotton ones [11]. This observation is applicable for all sets of oil-impregnated Kraft mock-ups, suggesting that the

density of the paper pulp did not allow the penetration and recess of the linseed oil to the same extent as the other two types of paper, providing an explanation for the differences in color and opacity changes (Sections 3.1 and 3.2).

### 3.4. Mechanical Strength Changes

Results have indicated that the conditions of artificial ageing caused limited alteration to the tensile strength of plain Montval and Kraft mock-ups (M and K) that presented a reduction of 9% at the final stages of ageing (Figures 14 and 15).

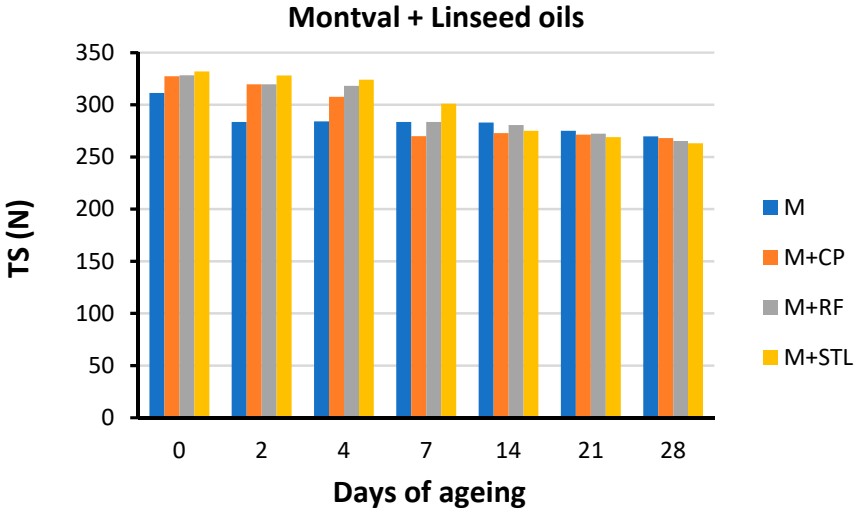

**Figure 14.** Changes of tensile strength of all sets of Montval mock-ups upon ageing. Ageing duration (0, 7, 14, 21, and 28 days) is shown on the horizontal axis.

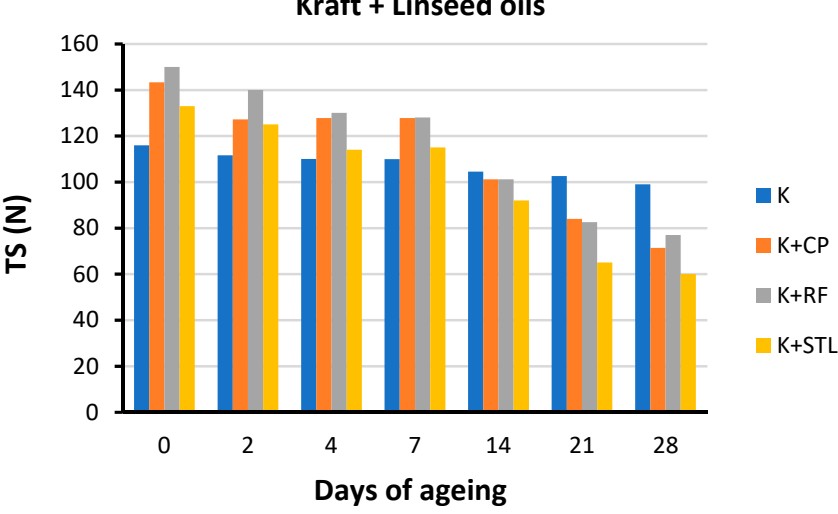

**Figure 15.** Changes of tensile strength of all sets of Kraft mock-ups upon ageing. Ageing duration (0, 7, 14, 21, and 28 days) is shown on the horizontal axis.

On the other hand, all sets of oil-impregnated Montval mock-ups presented a reduction of tensile strength upon ageing with no characteristic differences in the trend of changes between linseed oil formulations (Figure 14). Possibly, the thickness/weight of the paper, and thus the ratio of paper and oil volume, had an effect on that. After 40 days of drying, mock-ups presented an increase in tensile strength by an average of 7%, attributed to the application of linseed oils, which turn to a rubbery solid material after drying and constrain the fibres and pulp contents. The values remained almost stable up to the fourth day of ageing, then reduced gradually to those of the plain Montval mock-ups by the fourteenth

day, and they did not present noteworthy changes at the following stages. The values decreased up to an average of 20% (Figure 14) by the final stages of ageing.

The tensile strength measurements of all sets of oil-impregnated Kraft mock-ups presented a common trend of change in opacity with notable variations between linseed oil formulations (Figure 15). After 40 days of air drying, mock-ups presented an increase in tensile strength by 14% for those impregnated with stand-oil, 20% for those impregnated with cold-pressed linseed oil, and 23% for those impregnated with refined linseed oil, attributed to the application of linseed oils and their different properties. The values of the tensile strength of the mock-ups gradually reduced by an average of 13% on the fourth day of ageing, then reduced gradually to those of the plain Kraft mock-ups by the fourteenth day, while they decreased by an average of 50% in the final stages (Figure 15).

The above-mentioned changes refer to the results derived from the mock-ups cut parallel to the machine direction of the papers, Montval and Kraft. The mock-ups cut cross to the direction of the papers presented similar trends, respectively.

It could be suggested that the differences in reduction of tensile strength upon ageing depend both on linseed oil formulations and paper properties (weight) and pulp content. Differences between linseed oil formulations, attributed to their different properties, could be mainly observed in Kraft mock-ups (Figure 15), where paper weight is smaller. The limited reduction of oil-impregnated Montval mock-ups can be attributed to the sigificant weight difference between Kraft and cotton, and thus, the ratio of paper and oil volume. On the other hand, in comparing the results of oil-impregnated Kraft mock-ups with those of oil-impregnated mock-ups of cotton that have similar weight, there is a significant difference in tensile strength upon ageing, as cotton mock-ups were reduced by up to 80%. It could be suggested that differences in oil penetration–recession into the fiber net influenced the mechanical properties of the paper–oil system.

*3.5. Chemical Changes*

3.5.1. pH Measurements

The pH measurements showed that the application of linseed oil of all types causes a decrease in pH value of up to 2.4 points after 40 days of air drying. The mock-ups impregnated with cold-pressed linseed oil (M + CP and K + CP) presented a larger decrease (values range from 5.2 to 4.4 for M + CP and from 5.3 to 4.9 for K + CP), while the values of those impregnated with refined linseed oil (M + RF and K + RF) were quite close (values range from 5.8 to 4.9 for M + RF and 5.7 to 5.2 for K + RF) (Tables 1 and 2). The pH values of the Kraft mock-ups impregnated with stand-oil (K + StL) were comparable to those impregnated with refined linseed oil (M + RF and K + RF) (values range from 5.4 to 4.8 for K + StL), with those of Montval being slightly higher (values range from 6.0 to 5.6 for M + Stl) (Tables 1 and 2).

**Table 1.** pH measurements of all sets of Montval mock-ups.

| Days of Ageing | M | M + Cp | M + Rf | M + StL |
|:---:|:---:|:---:|:---:|:---:|
| 0 | 7.4 | 5.0 | 5.8 | 6.0 |
| 1 | 7.2 | 5.2 | 5.4 | 5.8 |
| 2 | 7.1 | 5.2 | 5.3 | 5.7 |
| 4 | 7.1 | 5.0 | 5.2 | 5.8 |
| 7 | 7.2 | 4.9 | 5.2 | 5.6 |
| 10 | 7.2 | 4.8 | 4.9 | 5.8 |
| 14 | 7.0 | 4.4 | 5.0 | 5.8 |
| 21 | 7.2 | 5.0 | 5.3 | 6.0 |
| 28 | 7.3 | 5.2 | 5.4 | 5.9 |

**Table 2.** pH measurements of all Kraft mock-ups.

| Days of Ageing | K | K + Cp | K + Rf | K + StL |
|:---:|:---:|:---:|:---:|:---:|
| 0 | 7.3 | 4.9 | 5.4 | 5.2 |
| 1 | 7.3 | 5.1 | 5.6 | 5.2 |
| 2 | 7.4 | 5.1 | 5.2 | 5.2 |
| 4 | 7.4 | 5.1 | 5.7 | 5.4 |
| 7 | 7.3 | 4.8 | 5.6 | 5.2 |
| 10 | 7.3 | 5.3 | 5.6 | 5.0 |
| 14 | 7.5 | 5.2 | 5.5 | 4.8 |
| 21 | 7.4 | 5.2 | 5.5 | 5.0 |
| 28 | 7.3 | 5.3 | 5.5 | 5.0 |

However, variations of pH values of all mock-ups was limited and variable (Tables 1 and 2) upon ageing. The same behavior has been noted for pure cotton mock-ups, although the decrease of the pH values was more intense, respectively. It could be suggested that the presence of alkaline buffer in Montval and Kraft papers played a role in the changes. The fact that oil-impregnated Montval mock-ups presented slightly higher values could be attributed to pulp processing and content. Generally, the results of pH values indicate that the application of linseed oil formulations establishes an acidic condition in the paper–oil system in the early stages of ageing.

3.5.2. FTIR Analysis

FTIR analysis showed a typical lignocellulosic paper profile, based on the results of previous works [21–25]. Wood pulp can be indicated by the lignin absorption at about 1730 (esters in hemicellulose fraction), 1590 (generally attributed to aromatics and possibly, to carboxylates), 1505 (aromatics, a lignin marker), 1450 (recorded in historic papers), ~1265 (broad absorbance due to C-O of guaiacyl ring of lignin residues), ~900 (glycosidic linkages in polysaccharide units), and 808 $cm^{-1}$ (typical of hemicelluloses). For the Montval paper, removal of lignin was indicated by the lack of all lignin markers, while it showed characteristic absorption at 1202–1204 $cm^{-1}$ (mainly due to the exocyclic $CH_2$ twisting of the glucose rings with contributions from other vibrations), 1050 $cm^{-1}$, and 1030 $cm^{-1}$ (due to various C-O vibrations of the polysaccharide structure) [25]. FTIR analysis additionally confirmed the presence of $CaCO_3$ in both papers by the marked absorptions at about 1430 and 874 $cm^{-1}$ along with clay, shown at 1030–1000 and 910 $cm^{-1}$, typical of aluminum silicates. Both materials were also detected by EDX analysis.

After 40 days of drying, FTIR spectrum of oil-impregnated mock-ups additionally showed the characteristic carbonyl band at 1745 $cm^{-1}$ and the C-O stretching pattern at 1239, 1164, and 1101 $cm^{-1}$, which is diagnostic for triglyceride ester linkages [25–28].

The typical bands due to rocking deformation of long alkyl chains at ~720 $cm^{-1}$ were observed in addition to the cis olefinic C-H bending vibration at 970–980 $cm^{-1}$ and the C=C stretching vibrations at 1680–1600 $cm^{-1}$, which are attributable to the conversion of unconjugated, disubstituted cis double bonds, as expected for the fatty ester in siccative oils, to trans upon drying [12,25,26,29–31].

As shown in Figures 16a and 17a, FTIR spectra of plain Montval and Kraft mock-ups did not display notable changes upon ageing. On the other hand, the spectra of oil-impregnated mock-ups showed moderate changes in specific bands of the spectrum upon ageing (Figure 16b–d for Montval mock-ups and Figure 17b–d for Kraft mock-ups). Stronger bands were observed due to the gradual intensity increase of the bands corresponding to carbonyl-containing species (such as aldehydes and ketones) and carboxyl acids (1600–1750 $cm^{-1}$), which are associated both with paper and linseed oil oxidation [23–26,29–32]. A marked increase was also noted in bands that correspond to oxidation compounds, such as hydroperoxides and alcohols (3200–3600 $cm^{-1}$ and 1100–1210 $cm^{-1}$) and the formation of conjugated bonds (such as in 1624, 1633, 950, and 723 $cm^{-1}$), a significant change that we attributed to the degradation of linseed oil,

confirming colorimetric measurements (see Section 3.1); additionally, oxidative polymerization in the final stages of ageing, mainly expressed through additional C-C bonding (1099–1238 cm$^{-1}$), is typically observed [25,26,29–31].

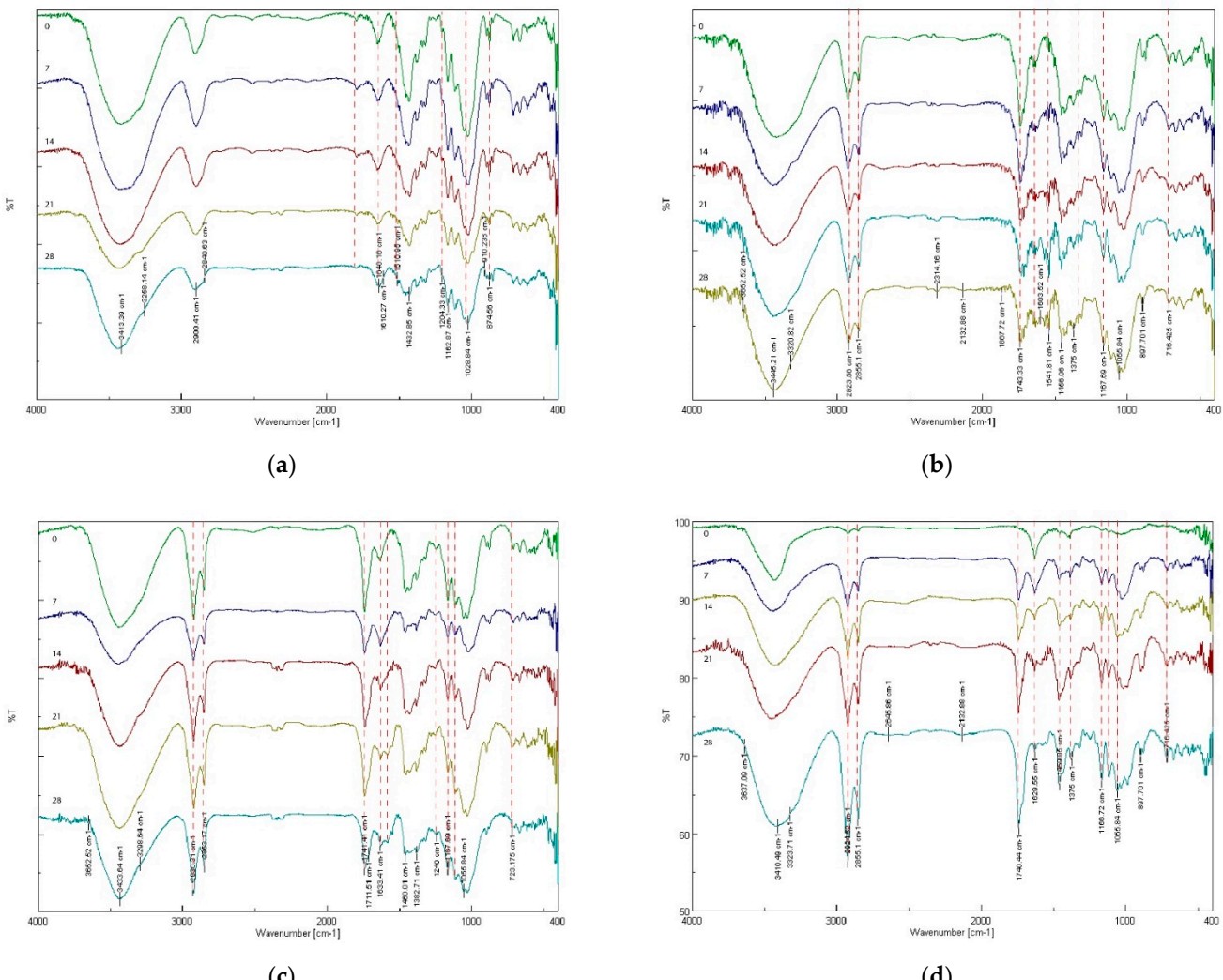

**Figure 16.** FTIR spectra of all sets of Montval mock-ups upon ageing: (**a**) plain one; (**b**) impregnated with cold-pressed linseed oil, M + CP; (**c**) impregnated with refined linseed oil, M + RF; (**d**) impregnated with stand-oil, M + Stl. Ageing duration (0, 7, 14, 21, and 28 days) is shown on each spectrum.

Variations in bands between the sets upon ageing are possibly associated with the manufacturing processes of linseed oil (Figure 16b–d for Montval mock-ups and Figure 17b–d for Kraft mock-ups). More intense changes are recorded to the mock-ups after the 14th day of ageing and more. The changes in paper chemistry could not so far be clearly determined due to band overlaps with those of linseed oil. Further study is required for the study of the changes in specific bands.

### 3.5.3. Analysis of VOCs through SPME-GC-MS

The analysis of volatile organic compounds (VOCs) provides indirect, yet critical information on degradation attributed to scission reactions induced during the ageing process to both oil and paper. The simultaneous ageing of these materials as a result of this specific experimental design offers an insight into the mutual impact between oxidized and hydrolyzed oil triglycerides and the cellulose and lignin of paper.

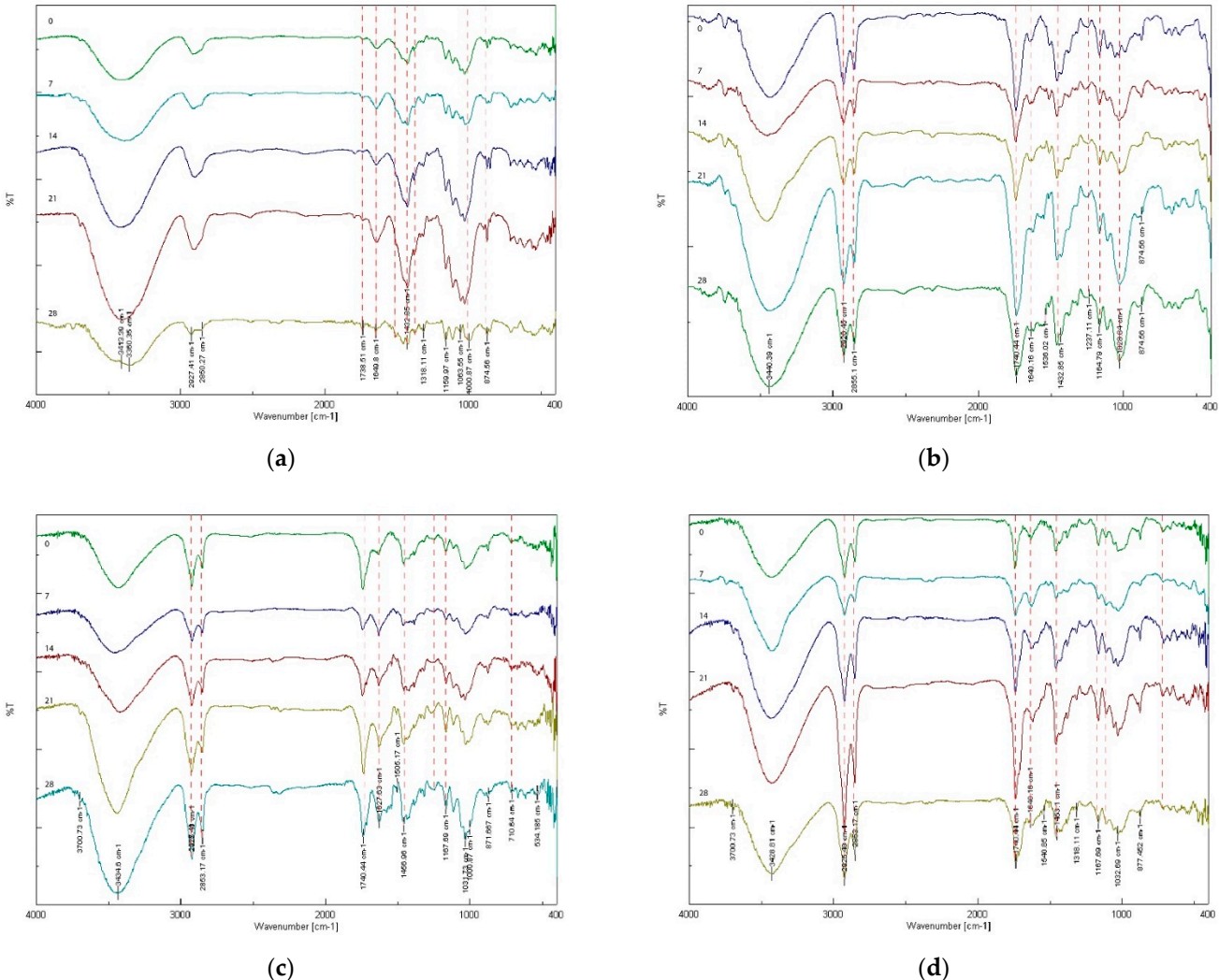

**Figure 17.** FTIR spectra of all sets of Kraft mock-ups upon ageing: (**a**) plain one; (**b**) impregnated with cold-pressed linseed oil, K + CP; (**c**) impregnated with refined linseed oil, K + RF; (**d**) impregnated with stand-oil, K + StL. Ageing duration (0, 7, 14, 21, and 28 days) is shown on each spectrum.

VOCs were sampled non-destructively, using Solid Phase Microextraction (SPME) and an SPME needle cartridge with a 50/30 with DVBCAR/PDMS (divinylbenzene–carboxen/poly(dimethylsiloxane)) fiber coating. The same methodology has been successfully applied for the investigation of the effect of linseed oils on pure cellulosic papers, following the equivalent process for original oil sketches and prints [11]. The results on mock-ups could be used as references for determining the stages of ageing or deterioration in original works on lignocellulosic papers.

GC-MS analysis of all sets of oil-impregnated Montval and Kraft mock-ups identified the same 30 compounds recorded for the oil-impregnated pure cellulosic papers with the three formulations of linseed oil [11]. These volatile organic compounds belong to several chemical classes: saturated and unsaturated aldehydes, ketones, alcohols, carboxylic acids, lactones, and furans. The carboxylic acids encompass methanoic (formic), ethanoic (acetic), propanoic, pentanoic, hexanoic, heptanoic, octanoic, and nonanoic acids. The aldehyde range of compounds comprises hexanal, 2-hexenal, heptenal, 2-heptanal, octanal, 2-octenal, nonanal, 2-nonenal, decanal, 2-decenal, and 2-undecanal. Ketones include 2-heptanone, 2-octanone, 4-nonanone, 2-nonanone, and 2-decanone. Lactones include γ-heptalactone and γ-nonalactone, furans include 2-pentyl furan, 5-ethyl-2(5H)-furanone and 5-penty-2-(5H)-furanone, while for alcohols, only 1-octen-3-ol.

These compounds have been associated with variant types of linseed oil, pure cellulosic and lignocellulosic papers [33–45], and their oxidation upon ageing. Saturated and unsaturated aldehydes, ketones, alcohols, carboxylic acids, and lactones have been mainly attributed to oil oxidation. It should be noted that the full range of C1-C18 acids is expected during oxidative degradation, however, the more volatile ones are detected with a specific type of chromatographic analysis (see Experimental). On the other hand, hexanal, heptanal, octanal, nonanal, and decanal have been detected in historical books with pure cellulosic fiber and wood pulp content [38], and they have been mainly attributed both to the lipid content of books and the natural ageing of paper [34,37]. Finally, furans mainly originate from the polysaccharide material in the paper.

To study the evolution of the emission of VOCs, the results were included in sum-up graphs for every chemical group: acids, aldehydes, ketones, and furans. For all three sets of Montval and Kraft oil-impregnated mock-ups, the emission of acids was comparatively higher than the other chemical groups and presented a common trend: a gradual decrease up to the 28th day of ageing (Figures 18a, 19a and 20a for Montval mock-ups and Figures 21a, 22a and 23a for Kraft mock-ups). However, at the final stages of ageing, their emissions remained at a higher level in comparison with other chemical groups. Among the three sets, both Montval and Kraft mock-ups impregnated with stand-oil present comparatively lower emission of acids and a more limited decrease (Figures 20a and 23a). This was also observed on the sets of cotton mock-ups [11], indicating that the type of oil influences the behavior. On the other hand, all sets impregnated with cold-pressed linseed oil presented a more intense decrease up to the 7th day of ageing. Formic, propanoic, and hexanoic acids presented higher emissions than the rest for all sets of mock-ups.

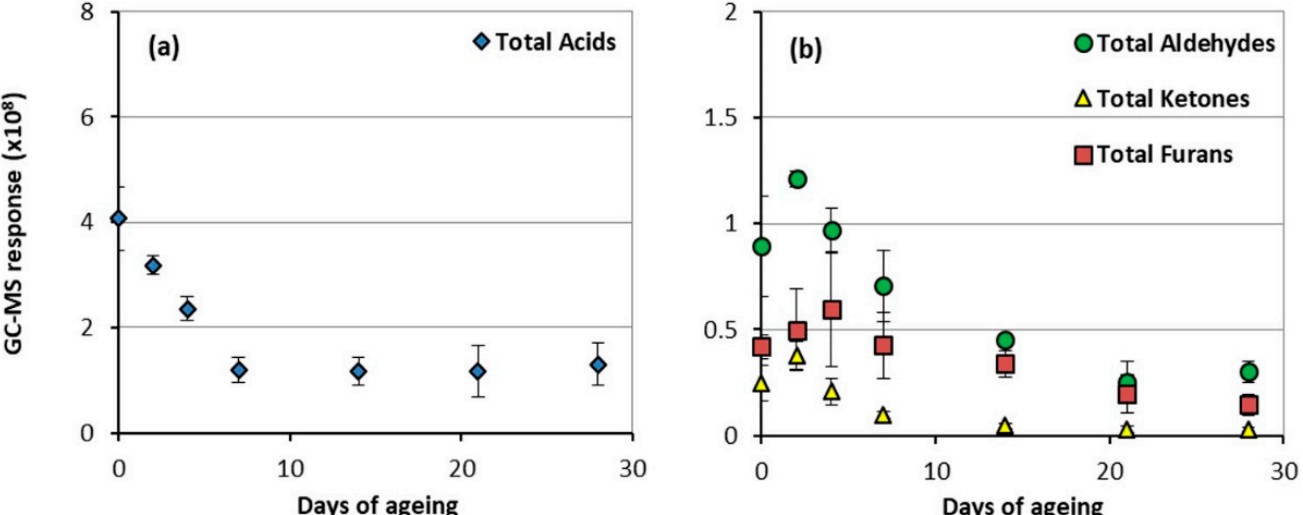

**Figure 18.** Sum-up evolution of VOCs belonging to a specific chemical group, emitted by Montval mock-ups impregnated with cold-pressed linseed oil upon ageing: (**a**) acids; (**b**) aldehydes, ketones, and furans.

Emissions of aldehydes were much lower than those of acids, and their trend, taking into consideration the standard deviation, appeared to be similar for the three sets of Montval and Kraft mock-ups. In the case of cold-pressed linseed oil, an increase up to the second day was detected for aldehydes, ketones, and furans, followed by a gradual decrease up to the 28th day of ageing (Figures 18b, 19b and 20b for Montval mock-ups and Figures 21b, 22b and 23b for Kraft mock-ups). This could be reasonable, since cold-pressed linseed oil is the least-treated oil, and thus is the more susceptible to oxidation. Aldehydes, ketones, and furans (secondary oxidation products of lipids) are being formed rapidly from the beginning of the accelerated oxidation, reaching a peak until the exhaustion of precursor compounds is reached, and then their emissions follow the same pattern as in the sets of refined linseed oil and stand-oil. It should be noted that the type of lignocellulosic

mock-up seems not to affect this phenomenon, while in the case of cotton mock-ups [11], aldehydes, ketones, and furans of cold-pressed oil followed a declining trend from the beginning of accelerated oxidation.

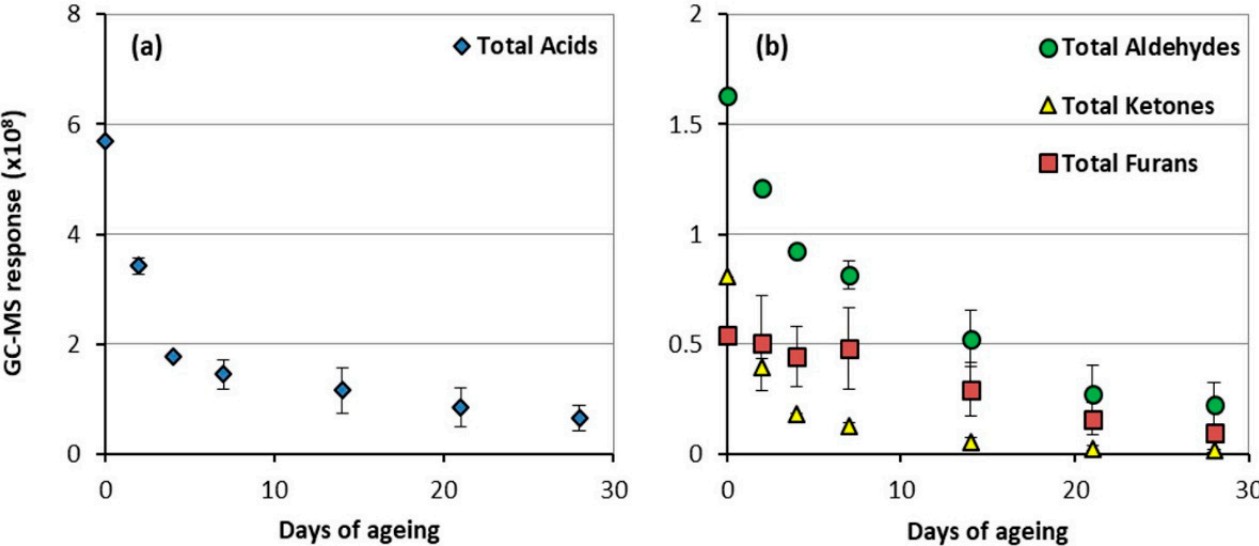

**Figure 19.** Sum-up evolution of VOCs belonging to a specific chemical group, emitted by Montval mock-ups impregnated with refined linseed oil upon ageing: (**a**) acids; (**b**) aldehydes, ketones, and furans.

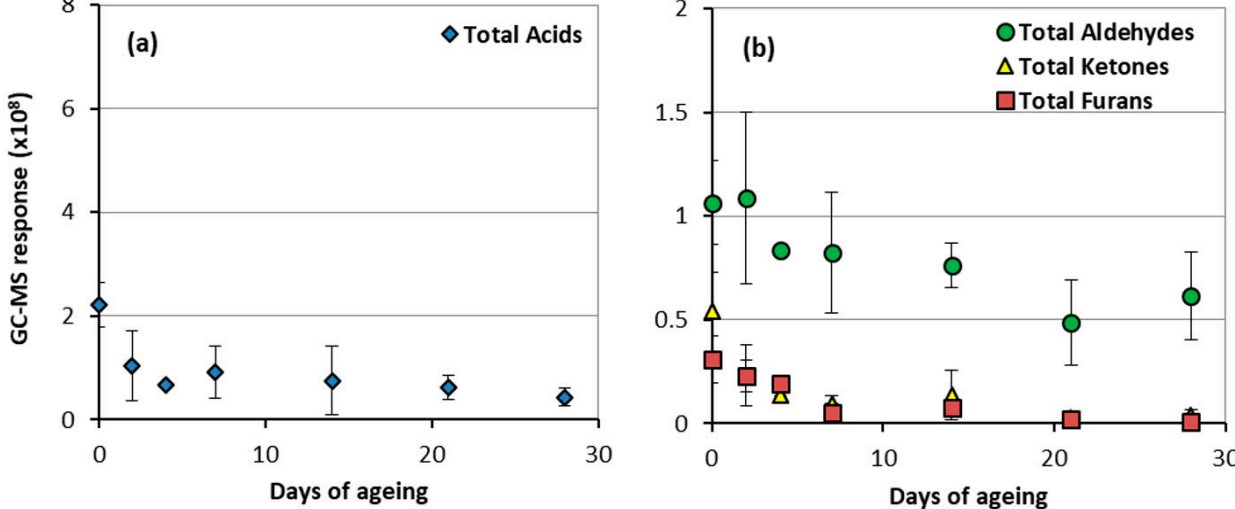

**Figure 20.** Sum-up evolution of VOCs belongin to a specific chemical group, emitted by Montval mock-ups impregnated with stand-oil upon ageing: (**a**) acids; (**b**) aldehydes, ketones, and furans.

Hexanal presented the highest values of emission for sets. For the mock-ups impregnated with stand-oil, the change of emissions upon ageing was comparatively limited, thus the total reduction was smaller (Figures 20b and 23b). Emissions of ketones and furans were lower than those of aldehydes for all sets of mock-ups. They presented a common trend of a gradual, limited, yet analogical decrease up to the 28th day of ageing (Figures 18b, 19b and 20b for Montval mock-ups and Figures 21b, 22b and 23b for Kraft mock-ups).

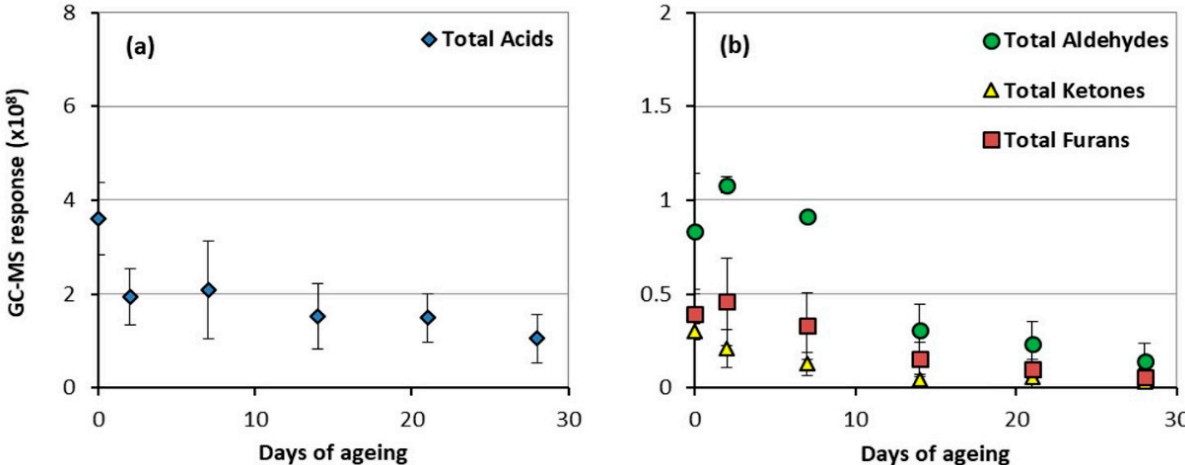

**Figure 21.** Sum-up evolution of VOCs belonging to the same chemocal group, emitted by Kraft mock-ups impregnated with cold-pressed linseed oil upon ageing,: (**a**) acids; (**b**) aldehydes, ketones, and furans.

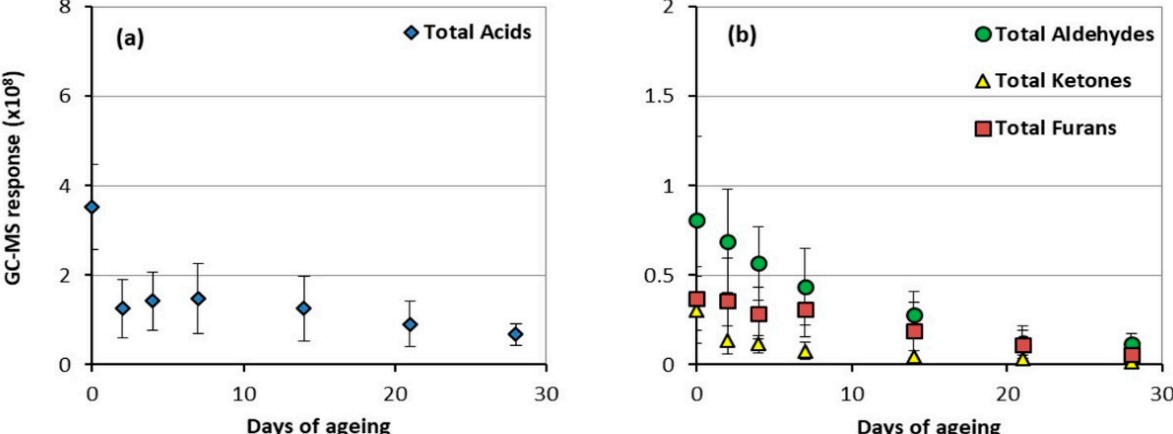

**Figure 22.** Sum-up evolution of VOCs belonging to a specific chemical group, emitted by Kraft mock-ups impregnated with refined linseed oil upon ageing: (**a**) acids; (**b**) aldehydes, ketones, and furans.

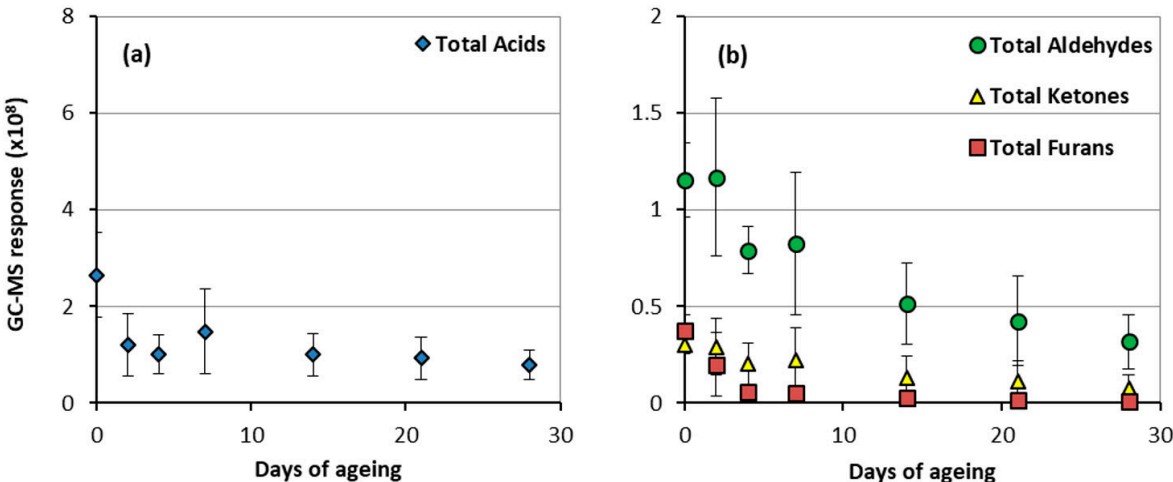

**Figure 23.** Sum-up evolution of VOCs belonging to a specific chemical group emitted by Kraft mock-ups impregnated with stand-oil upon ageing: (**a**) acids; (**b**) aldehydes, ketones, and furans.

Finally, the emissions of lactones and alcohol were significantly low and presented a similar trend for all sets of mock-ups. In particular, lactones presented an increase of emissions up to the 7th day of ageing and then a gradual fall, while the 1-octen-3-ol presented a gradual fall up to the 14th day and then flattened.

As for pure cellulosic papers, the trend of emissions of the several chemical groups could be mainly associated with the kinetics involved with the drying and degradation of linseed oils [41,42]. The extensive oxidation of linseed oil and the consequent polymerization/degradation gradually inhibits the emission of volatile organic compounds upon ageing [11].

Also, it has been indicated that the process of manufacture of each formulation of linseed oil influences the quantity and trend of emission, as for pure cotton mock-ups [11]. Stand-oil, as mentioned already, is a pre-polymerized oil for which the process of drying is slower than in the cases of cold-pressed and alkali-refined linseed oils [13].

Therefore, the three-dimensional matrix of polymerization is formed at a slower rate and allows the light volatile compounds, like aldehydes (e.g. boiling point of hexanal is 129 °C), to be emitted for longer periods in the headspace of the mock-up. This phenomenon has been observed mainly for aldehydes in all types of paper, i.e., cotton [11], Montval and Kraft, but it seems that the type of paper also influences the intensity of the phenomenon. The rate of decline of the emissions of aldehydes was more intense for Kraft mock-ups, in comparison to that of cotton and Montval mock-ups that presented similar behavior. It could be suggested that the stand-oil film, which remains on the surface of Kraft mock-ups up to the final stages of ageing without recessing into the fiber net, might influence the emissions of the system. The emissions could mainly derive from the oil film.

Further work is required for the evaluation of the results. The processing of emission results of every compound is regarded as necessary to export data useful for the condition assessment of oil-impregnated paper areas on original works of art, hopefully enabling the determination of markers that indicate the condition of oiled areas of a paper support.

## 4. Conclusions

The results of this work contribute to the research on the effect of linseed oils on paper support. Investigation on lignocellulosic papers contributes to the previous results on pure cellulosic paper beyond the effect of the provenance and processing of the fibers, with factors like color, paper weight and paper: oil volume ratio, pulp content, and oil penetration that influence the changes recorded over time.

Application of linseed oils is mainly responsible for the trend of all types of changes on mock-ups, while variations between the sets of each paper type could be attributed to the differences in oil processing during manufacture and the resultant properties respectively. The differences between the sets of the two papers sets seem to depend on the penetration and recess of the oil into the fiber net, which is associated with pulp content and density, and the paper weight, and thus the ratio of paper and oil volume. This was evident in the results of the color and opacity measurements of the sets.

The extent and the rate of change of oil-impregnated Kraft mock-ups upon ageing were comparatively limited, probably due to its dense paper pulp preventing oil penetration and recess. Moreover, the trends of optical changes of oil-impregnated Montval mock-ups were comparable with those of pure cellulosic–cotton papers [11] (both white), with notable variations that indicate that paperweight and ratio of paper: oil volume might have influenced the outcome. The reduction of tensile strength for the two papers differs, with oil-impregnated Montval mock-ups presenting comparatively smaller changes. Taking into consideration the collapse of the tensile strength of the oil-impregnated pure cellulosic–cotton mocks, it could be suggested that the presence of alkaline buffer might restrain the chemical deterioration of the support. In addition, pure cotton paper had no sizing, fillers, or additives, so linseed oil formulations penetrated through the fibre net unobstructed and consequently had a major effect on the paper–oil system. FTIR spectra of oil-impregnated cotton mock-ups also indicated more extended chemical changes at the bands of peroxides,

aldehydes, ketones, and carboxylic acids [11], indicating hydrolysis of linseed oil and the consequent acidic hydrolysis of paper. On the other hand, Montval's weight is 80% higher than that of Kraft and 100% more than that of cotton paper. This could set a hypothesis, to be investigated, that the ratio of oil and paper mass could have an effect on the mechanical behavior.

Generally, the chemical changes presented in the FTIR analysis support the changes recorded for the sets of mock-ups, with other methods of study providing possible explanations for the changes recorded. VOC's emissions appear to be influenced mainly by linseed oil processing. The trend of emissions presents a gradual decrease upon ageing, remaining comparatively low after the 14th day of ageing in response to the respective chemical changes. The input of paper type can be noted in the differences in the trend of VOC emissions between cotton, Montval and Kraft oil-impregnated mock-ups.

Finally, a comparative study of the results of this work could not point out specific stages of deterioration upon ageing yet. It could be suggested that the changes on the mock-ups can be divided into three phases: the first, from 0 to 4 or 7 days; the second, from 4 or 7 to 14 days; and the third, from 14 to 28 days. The research will proceed with advanced statistical processing of the results to establish the stages of deterioration with mathematical tools. This will compensate for the condition assessment of works created with oil media on paper, thus aiding in conservation and preservation decision-making for a wide range of cultural heritage objects.

**Author Contributions:** Conceptualization, P.B.; methodology, P.B., A.A., C.T., D.T., L.-A.T. and T.K.; software, P.B. and D.T.; validation, P.B., A.A., C.T., S.B., D.T., L.-A.T. and T.K.; formal analysis, P.B., A.A., S.B., D.T., L.-A.T.; investigation, P.B.; resources, A.A.; data curation, P.B.; writing—original draft preparation, P.B.; writing—review and editing, P.B., S.B., D.T., L.-A.T. and A.A.; visualization, P.B. and D.T.; supervision, A.A., C.T. and K.C.; All authors have read and agreed to the published version of the manuscript.

**Funding:** This work was performed in the framework of a PhD research project, and it has not received any funds from external sources.

**Informed Consent Statement:** Not applicable.

**Data Availability Statement:** Not applicable.

**Conflicts of Interest:** The authors declare no conflict of interest.

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
