# Peer review of "Oil Media on Paper: Investigating the Effect of Linseed Oils on Lignocellulosic Paper Supports"

_analytica, doi:10.3390/analytica3030019_

Round 1
Reviewer 1 Report
The paper could be of high interest to reader if it would gain more scientific discussion.

Author Response
Dear reviewer,
We would like to thank you for your comments and remarks. We have tried to respond to them and make the necessary changes. This really helped the improvement of the paper's content. Please, find our answers to your questions about methodology in the attached file.
Kind regards,
Penelope Banou

Reviewer 2 Report
The article is well written, and analytically executed.
A little challenge is to separate the oil degradation (could be done on ageing on glass), to separate the effects from the paper. As papers are quite different in grammage an type, the obtained data is little limited. This is obvious from Figure 2 and 3. where porosity and absorbtion differences show big differences in spectra (connected with the depth of oil penetration). This is a suggestion for further studies,
Author Response
Dear reviewer,
I would like to thank you for your comments and suggestions. I wish that soon we will be able to publish the results of the investigation on the three linseed oil formulations (exclusively). The parameter of paper type is also another matter to consider as our research goes on.
Kind regards,
Penelope Banou
Reviewer 3 Report
In this article authors have performed many experiments to establish their findings and studies to Investigate the Effect of Linseed Oils on Lignocellulosic Paper Supports. Manuscript was written in a systematic way which is very understandable by the reader. Therefore I like to request editor to accept the manuscript for publication as it does perfectly goes with the aims and objectives of this journal. My few minor corrections are below.
1. How they quantify the individual peaks, have they run any standard? which is not clear. it would better if they clearly explain that portion.
2. if the run standard, are they run all standard together or individually?
Author Response
Dear reviewer,
We would like to thank you for your comments. We have tried to respond to your questions as follows:
For each identified compound by GC-MS, we selected one ion, the most abundant and unique in relation to the ions of the neighboring peaks (to avoid any partial overlapping) and this ion was used for the quantification of the individual peak. We identified 8 volatile carboxylic acids two of them having different quantification ions than the others (formic m/z=46, propanoic m/z=74, acetic, pentanoic, hexanoic, heptanoic, octanoic, nonanoic m/z=60), while in the case of aldehydes it is more complicated since almost for each of most compounds a different quantification ion was assigned. Therefore using a single standard for acids, a second one for aldehydes etc, could not serve as measures for total compounds of each group. The values that would have been produced, could only serve as conventional measures with low correspondence to reality. In our opinion the only way of quantifying individual compounds and groups of compounds is the construction of 30 calibration curves with the respective standards, which would have been quite difficult, at least at the current stage of research. Our approach was to sum up the individual quantification ions for each group of compounds, presented as “GC-MS response”. Our observations and findings concerning a specific group of compounds are directly comparable through different experiments.
The ultimate goal of our research which is in progress, is the development of a mathematical model that could describe the whole phenomenon of ageing, as mentioned above. We believe that from the 30 identified compounds we will be able of refining and detecting the ones that could serve as markers of ageing. Then we will develop the necessary calibration curves to directly quantify the compounds-markers.
No. |
Compounds |
r.t. (min) |
Quantification Ion (m/z) |
Identification Ions (m/z) |
4 |
Hexanal |
4.03 |
56 |
44, 72, 82 |
5 |
2-hexenal |
5.36 |
55 |
69, 83, 98 |
7 |
Heptanal |
6.88 |
70 |
55, 81, 96 |
9 |
2-heptenal |
8.40 |
83 |
55, 70, 112 |
13 |
Octanal |
9.38 |
84 |
69, 100, 57 |
16 |
2-octenal |
10.30 |
83 |
55, 70, 108 |
19 |
Nonanal |
10.97 |
57 |
41, 70, 98 |
22 |
2-nonenal |
11.68 |
41 |
70, 83, 96 |
25 |
Decanal |
12.21 |
57 |
82, 112, 128 |
26 |
2-decenal |
12.82 |
41 |
70, 83, 98 |
29 |
2-undecanal |
13.82 |
70 |
57, 83, 124 |
Kind regards,
Penelope Banou